# Nuclear Melt Glass from Experimental Field, Semipalatinsk Test Site

**Irina E. Vlasova** [1], **Vasily O. Yapaskurt** [2], **Alexei A. Averin** [3], **Oleg E. Melnik** [4], **Denis A. Zolotov** [5], **Roman A. Senin** [6], **Tatiana R. Poliakova** [1], **Iurii M. Nevolin** [1,3], **Stepan N. Kalmykov** [1] **and Andrey A. Shiryaev** [1,3,7,*]

1   Department of Chemistry, Lomonosov Moscow State University, Leninskie Gory 1 bld. 3, 119991 Moscow, Russia
2   Department of Geology, Lomonosov Moscow State University, Leninskie Gory 1, 119991 Moscow, Russia
3   Frumkin Institute of Physical Chemistry and Electrochemistry RAS, Leninsky Prospect 31 bld. 4, 119071 Moscow, Russia
4   Institute of Mechanics, Lomonosov Moscow State University, Michurinskii pr., 1, 119192 Moscow, Russia
5   FSRC "Crystallography and Photonics" RAS, Leninsky pr. 53, 119333 Moscow, Russia
6   National Research Center "Kurchatov Institute", Ak.Kurchatov sq., 1, 123182 Moscow, Russia
7   Institute of Geology of Ore Deposits, Petrography, Mineralogy and Geochemistry RAS, Staromonetnii per. 35, 119017 Moscow, Russia
*   Correspondence: a_shiryaev@mail.ru or shiryaev@phyche.ac.ru

**Abstract:** Investigation of shocked materials provides unique information about behavior of substances in extreme thermodynamic conditions. Near surface nuclear tests have induced multiple transformations of affected soils. Examination of nuclear glasses and relics of entrapped minerals provides a unique database on their behavior under an intense temperature flash. In this work, several types of nuclear fallout particles from historic tests at the Semipalatinsk test site are investigated using complementary analytical methods. Distribution of radionuclides in all types of samples is highly heterogeneous; domains with high content of radionuclides are often intermixed with non-active materials. There is no general correlation between chemical composition of the glassy matrix and content of radionuclides. In aerodynamic fallout, the main fraction of radionuclides is trapped in the outer glassy shell. Relics of quartz grains are always devoid of radionuclides, while glass regions of high activity have different composition. In contrast to underground tests, iron-rich minerals are not necessarily radioactive. In most cases, the glassy matrix in anhydrous and is strongly polymerized, and the $Q^3$ silicate groups dominate. Temperature-induced transformations of entrapped minerals are discussed. Investigation of zircon grains shows absence of a direct correlation between degree of decomposition into constituting oxides, morphology of resulting baddeleyite, and maximum experienced temperature. For the first time, temperature history of a nuclear ground glass is estimated from Zr diffusion profiles from decomposing zircon grain.

**Keywords:** nuclear fallout; nuclear melt glass; autoradiography; Semipalatinsk test site; zircon decomposition; high-temperature metamorphism

## 1. Introduction

Near-surface detonations of nuclear weapons lead to the formation of aerodynamic glassy particles, essentially, splashes of molten material, and of glassy crust beneath the epicenter ("Ground Zero"). These materials have been produced by intense radiation in a wide energy range and a shock wave, and comprised of by-products of vaporization/melting and quenching of test ground soils as well as test-related materials entrapped into a fireball. Early characterization works were mainly devoted to dose calculation, physical description of vaporization –condensation and droplets formation, and general classification of the glass materials [1–6]. In recent decades, the main research emphasis

shifted towards comprehensive determination of p-T and neutron flux conditions, source definition, speciation of Fe, U, and Pu in different types of explosions, reconstruction of the mechanisms in fallout formation, explosion yield, distance from the epicenter, etc. [7–11]. Several studies address structure, devitrification processes, and relationships between composition/structure/quenching and formation conditions of nuclear melt glasses [12,13]. These works complement results obtained for other types of materials formed during transient high temperature events, firstly, fulgurites and impactites.

Several broad groups of glassy materials formed during near-surface explosions are distinguished. One of the important varieties is a glassy radioactive crust enveloping pristine particles of the Ground Zero soil and covering former pools of molten material [2,3,7,14–21]. Another variety consists of aerodynamic glassy fallout particles represented by drops up to several millimeters in size; sometimes, several drops merge together [3,7,14,15]. These particles both form during mid-air condensation of vaporized materials and when the mixture of the vaporized device and molten local soil quenches upon touching the ground. As suggested by analysis of various isotopic systems, the formation of individual fallout particles and their radioactivity depends on the yield and altitude of the explosion and may last for up to several tens of seconds [5,17,22]. Examination of these materials provides information about oxygen fugacity ($fO_2$) of the fireball and first models suggested rather reducing the environment [8,22]. However, more recent studies show that the redox conditions are more heterogeneous and at least some fallout particles record changes in $fO_2$ on their flight trajectories [23,24].

The most comprehensively studied ground nuclear glasses are the so-called "trinitites" from the Ground Zero of the first nuclear explosion (the "Trinity" test) in Alamogordo, United States, in 1945 [2–4,7,12,16–20]. Knowledge of the exact time and yield of the detonation facilitate interpretation of the analytical studies and the development of methodological approaches for nuclear forensics [19,20]. Two main scenarios of ground melt glass formation are discussed: (a) melt droplets from the fireball falling onto the ground and spreading/melting over the local sands, and (b) direct melting of local sands by intense radiation. The former model finds support in recent studies [2], although there is little doubt that depending on particular sampling locality in Ground Zero, nuclear melt glasses from a single event may record a combination of both mechanisms. Sharp temperature rise was followed by slower cooling, allowing crystallization of individual mineral phases. In particular, iron-rich nuclear melt glass separates into two immiscible liquids and forms crystals of Fe oxides with various textures that carry information about the conditions of the glass quenching [25].

In this work, we present results of a complementary study on the radioactivity distribution, composition, and structure of the nuclear melt ground glass and aerodynamic glassy fallout formed during historical ground tests on the territory of the Experimental Field, Semipalatinsk test site, Republic of Kazahkhstan, former Soviet Union. The Experimental Field (Opytnoe Pole) of the Semipalatinsk test site (STS), is the locality of the first Soviet nuclear test in 1949, and the first Soviet thermonuclear test in 1953. In total, 30 atmospheric detonations of different yield and degree of success were conducted from 1949 to 1962. Only a few studies on fallout material from the STS were published [14,26–32]. The samples vary in specific $\alpha$-activity, thus reflecting different proportions of the device material and of local soils. The nuclear melt glasses are kinetically stable for several decades and effectively retain radionuclides: more than 30 years after the termination of the ground tests >90% of Pu and U in soils remain in acid-resistant residues [27–29]; the leach rates of actinides from glasses produced in underground tests at the STS are also low [30]. Since there has been a moratorium on ground and atmospheric tests effective since 1963, the major fraction of radioactivity in fallout plumes at the Experimental Field is concentrated in coarse soil fraction, largely in so-called «hot» particles [31]. Distribution of $^{239,240}$Pu in soil fractions and concentration of the "hot" particles per gram vary significantly in different sampling points of the Experimental Field due to different sources of the plumes [32].

## 2. Materials and Methods

### 2.1. Samples

Glassy melt fragments produced in historical nuclear tests were collected at the technical area P1 of the Experimental Field (Opytnoe Pole, Semipalatinsk test site) at different periods after the implementation of the Partial Test Ban Treaty in 1963. The particles were initially identified from their unusual appearance, markedly different from rocks common for the area: all the samples possess dark (black) glassy outer surface. All discussed samples are clearly considerably larger than local sand grains. With a certain degree of probability, the samples can be attributed to Ground Zero of the first Soviet explosion (19 August 1949; 22 kt), but precise timing and yield of the explosions is impossible since the same area was used for multiple nuclear (for instance, on 24 September 1951; 38 kt) and thermonuclear (August 1953; 400 kt) tests. Based on their morphological features, the samples can be divided into three groups (Figure 1).

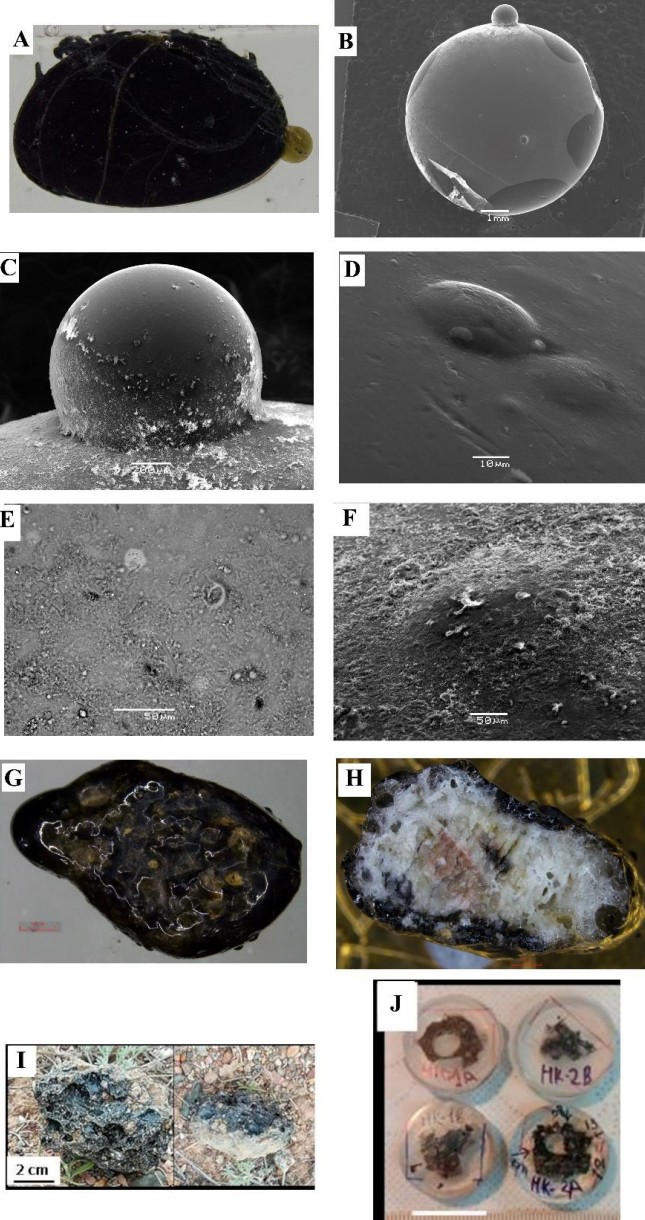

**Figure 1.** Appearance of the samples of the three studied groups. Group I—(**A–E**); Group II—(**G,H**); Group III—(**I,J**). (**A**) Optical image of a polished half of the sample R1 embedded in epoxy resin (1 mm); (**B**) SEM-SE images of R1: general view (scale bar 1 mm); (**C**) enlarged image of the small bead

on the surface (200 μm); (**D**) smaller beads (10 μm); (**E,F**) (50 μm) features of high-temperature etching, note the boundary between the etched and smooth surface in (**E**); (**G,H**) optical images of particles of the Group II ("kharitonka"); (**G**) outer appearance; (**H**) polished cross-section; (**I**) optical photographs example of top and side view of the black ground glass samples (Group III, HK); (**J**) epoxy resin tablets with embedded fragments of the "HK" samples.

Group I: Aerodynamic glassy fallout particles represented by specimen #R1 (Figure 1A–F). The sample is a fully vitrified black droplet with a glassy luster, shaped like a flattened sphere ~1 cm in diameter. Several small (<1 mm) glassy beads are attached to the surface. Both the largest droplet as well as the attached beads are decorated by yet smaller spheres with sizes ranging from sub-μm to ~10 μm. These smaller beads are usually deeply embedded into the carrier (Figure 1D,E). The surface of the sample is generally smooth, but in some extended rough domains are observed (Figure 1E,F). The roughness might represent devitrification of the glassy matrix, corrosion in hot environment of the fireball, or, although less likely, alteration in the environment.

Group II: Aerodynamic glassy fallout particles of the second type—individual fragments of elongated, rounded, or irregular shape, 0.5–2 cm in the largest dimension (Figure 1G,H). Small glassy beads adhered to the surface. The particles are entirely covered with black glass crust; the interior part consists of sintered sand. A colloquial Russian term for such samples is "Khariton" glasses (Yu.B. Khariton was a prominent physicist, the leader of Soviet nuclear project). Particles from this Group are denoted as Kh#.

Group III: Nuclear melt ground glass—layered samples of a glassy crust with abundant bubbles in the upper layer and unaltered sand in the lower layer; analogue of "trinitites" (Figure 1I,J). The samples are denoted as HK.

### 2.2. Methods

Morphology and composition of the samples were revealed using a JEOL JSM IT-500 scanning electron microscope equipped with an X-Max-50 energy dispersive spectrometer (Oxford Instruments). An acceleration voltage of 20 kV and a current of 0.7 nA was used. Quantification was performed using a set of reference materials; full ZAF correction was employed. Electron back-scattering diffraction (EBSD) was employed for determination of structure of selected crystalline phases.

Raman spectra were recorded using a 1200 lines/mm grating using inVia Reflex (Renishaw) and Senterra (Bruker) spectrometers equipped with Peltier-cooled CCD detectors. Spectra were acquired using $50\times$ and $100\times$ objectives with NA of 0.5 and 0.8, respectively. Accuracy and resolution of lines in the Raman spectra are $\pm 1$ cm$^{-1}$ and $\pm 2$ cm$^{-1}$, respectively. Laser power on the sample did not exceed 0.2 mW as measured using a LaserCheck device (Coherent); repeated control of spectra consistency was performed. Wavenumber calibration was performed using a Si reference sample and a high-quality CVD diamond. The spot size was 2–5 μm depending on magnification used.

Infra-red spectra in reflectance geometry were collected using an AutoImage microscope attached to a SpectrumOne FTIR spectrometer (Perkin Elmer), in the range from 600 to 5000 cm$^{-1}$; with apertures between 50 and 100 μm. Spectral resolution was 2 cm$^{-1}$.

X-ray tomography was performed at a RT-MT beamline at the Kurchatov synchrotron source. Monochromatic radiation from a bending magnet with energy of 20 keV was used. Spatial resolution of the detector is 10 μm. Treatment of the reconstructed data set mostly consisted of threshold filtration to highlight contribution of pores/bubbles and of dense mineral phases; the segmentation process is similar to that described in [33].

X-ray fluorescence (XRF) measurements were performed using Bruker Tornado M4 Plus spectrometer. X-ray tube with Rh anode operating at the acceleration voltage of 50 kV and a current of 0.2 mA was used. Polycapillary optics focuses the incident beam down to ~20 μm. Note that the spot size is the function of incident photon energy and increases

for soft radiation. The samples were evacuated to ~1 Pa. The calculation of the content of elements was performed using a fundamental parameters approach.

High purity Ge (HPGe) ORTEC detector with SpectraLine software was used for γ-spectrometry. Gross α/β-radioactivity was counted using low activity radiometer UMF-2000 (Doza Ltd., Zelenograd, Russia). The calculation of specific activity (per gram) was used for comparison of the radioactivity of the glass fragments. It should be noted that the ratio of the total activity to the sample mass is not necessarily a good measure of the content of radionuclides in the active part of the fragments. In some samples, radioactivity is fairly homogeneously distributed, but in the particles of the second group the activity, it is concentrated in the outer layer only. Due to a different ratio of the active and inactive parts of the glass fragments, the specific γ- activity and especially α- and β- activity decreases for larger particles due to larger contribution of barren core.

A polycarbonate track detector (TASTRAK, Bristol, UK) and optical microscope Olympus BX-51 with ImageScopeM software were used for acquisition of α-track images. Imaging plate radiography was performed using Cyclone Plus Storage Phosphor System (PerkinElmer).

## 3. Results

### 3.1. Gross α/β-Radioactivity and γ-Spectrometry of the Samples

Gross α-activity data allowed the three distinct groups of the melt glasses to be distinguished (Figure 2A). The total α-activity follows the sequence: R1 (Group I) > "Khariton" glasses (Group II) > ground glasses (Group III). At the same time, the specific β-activity of samples from groups II and III is similar and lower than that of Group I.

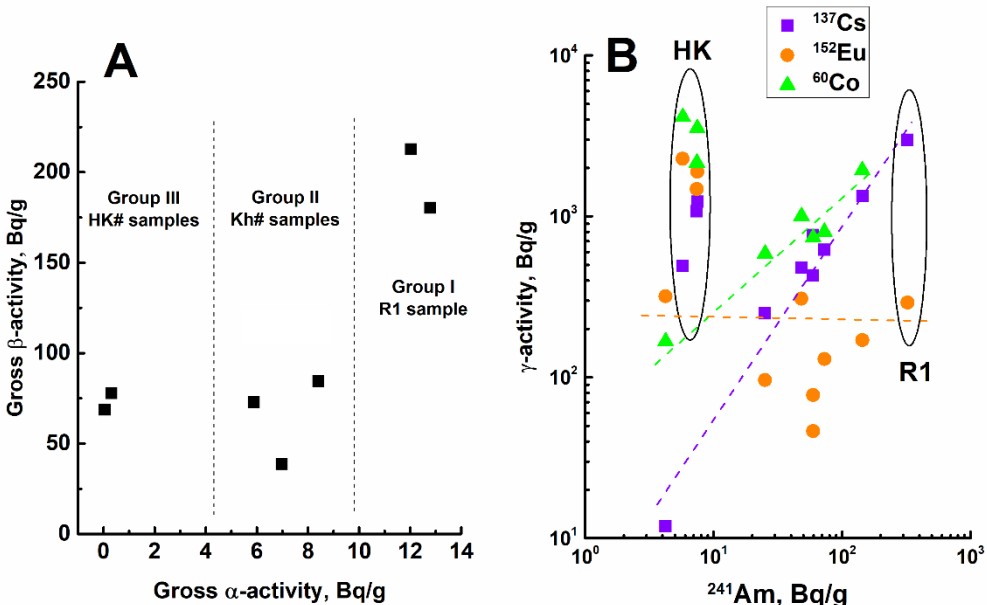

**Figure 2.** Radioactivity of the particles. (**A**) Gross α- and β- activity; vertical dashed lines separate different groups of the samples (see text); (**B**) γ-spectroscopy data of three groups of particles: the circle marked as "R1" shows Group I; the circle HK is related to Group III; all other points belong to Group II ("Khariton" particles). The lines are just to guide the eye.

Of the four γ-photon emitters that are confidently detected in the nuclear glass samples, $^{241}$Am is a clear marker of the device material contribution. The specific $^{241}$Am activity follows the order: Group III < Group II < Group I (Figure 2B). A large particle Kh-2 from Group II is exceptional, since its $^{241}$Am specific activity is even lower than in the ground crusts from Group III. This low value can be explained by large particle size, and, accordingly, the largest contribution of inactive central core. In the Group II particles, the radioactivity is concentrated in a thin glassy outer shell. The shell thickness is rather

constant and is independent on the specimen size, which results in a large spread of the specific activity values in the Group II particles.

Data for fission/activation products ([137]Cs, [152]Eu and [60]Co) are shown in Figure 2B as a function of [241]Am specific activity. $\gamma$-spectroscopy data for [137]Cs, [60]Co, and [152]Eu were recalculated to the date of the first candidate explosion in August 1949. This assumption is based on the sampling location coinciding with Ground Zero of the first nuclear bomb test; note that contribution of material from other tests (not later than 1962) cannot be excluded. Recalculation for another date, e.g., for August 1953, reduces the activity of [60]Co by 40%, [152]Eu by 20%, and [137]Cs by 10% relative to the assumed test date in August 1949. The [137]Cs activity varies significantly. The maximum value of $2.7 \times 10^3$ Bq/g for the R1 particle (Group I) is 2–10 times higher than the [137]Cs values for particles of the other two Groups. The activities of fission and activation products as well as the [241]Am activity in particles of Group I (R1) and Group II ("Khariton" samples) follow a common trend: it depends on the relative mass fraction of the active glassy part. The fully vitrified particle R1 (Group I) is the limiting example. The ground glasses of Group III (HK) represent a distinct group with the highest specific activity of activation products [152]Eu and [60]Co.

### 3.2. Autoemission Spectra of Actinides

Emission of characteristic X-ray photons by "hot" recoil ions producing an $\alpha$-decay event is a well-known phenomenon in actinide science. In samples with substantial amounts of high specific activity isotopes the intensity of these emissions may even compromise quantitative EDX analysis [34]. In addition, interaction of $\alpha$-particles with the matrix induces X-ray emission (particle-induced X-ray emission, PIXE), providing information about its elemental composition. Despite low gross $\alpha$-activity of the samples studied in this work (Figure 2), we have attempted to observe autoemission of the $\alpha$-decay products. The EDX spectrum of the R1 sample was measured for 80,000 s with the electron beam switched off, see Figure 3. Note that qualitatively similar results are also obtained for some other samples (e.g., for Kh-7), although lower exposure precludes analysis.

Despite poor statistics, it is possible to assign the observed features to known transitions, although tentatively. Strong background at the smallest energies likely represents fluorescence from O in constituent oxides and C from the coating. In the low energy range, PIXE contribution dominates and peaks due to Na and/or Mg (1.19 keV), Si (1.73 keV), and Fe (6.4 and 7.06 keV) are present. Strong peaks in the high energy range (13–20 keV) are confidently assigned to L-lines of U and Np. In the studied sample, these elements are recoils from $\alpha$-decay of Pu and Am isotopes. It is tempting to assign the peak at 15.8 keV to Zr K$\alpha$ excited by $\alpha$-particles. However, concentration of this element in the glassy matrix sample is generally below detection limits of the employed EDX spectrometer. Although grains of Zr-rich minerals (zircon and baddeleyite, see Section 3.5.2) are present in the samples, their volumetric fraction is insignificant and the appearance of the strong Zr K$\alpha$ line would require unrealistically high concentration of $\alpha$-sources in their vicinity. Moreover, cross-sections of ($\alpha$, Zr) reactions are very small in the relevant energy of incident particles from actinide decay (less than 5.5 MeV) even for low abundance [96]Zr, which readily forms under high neutron fluxes expected for nuclear tests.

Consequently, we assigned the peak at ~15.8 keV to an unusually strong U L$_{\beta 6}$ line (the L$_3 \rightarrow$N$_1$ transition) with eventual contribution of some other transitions. The enhancement of this line presumably results from several processes accompanying passage of a "hot" recoil ion through the matrix. Excited heavy recoil nucleus (in our case U or Np) may eject inner-shell electron ("internal conversion" [34]). This process is further complicated by interpenetration of electron shells of a "hot" and lattice ions ("electron promotion" [35]) and by the Coster–Krönig effect [36]. Consequently, the intensity of the X-ray emission lines produced by the recoil ions may differ notably between various matrices, since the chemical environment of the emitting ions varies (see, e.g., discussion in [37]).

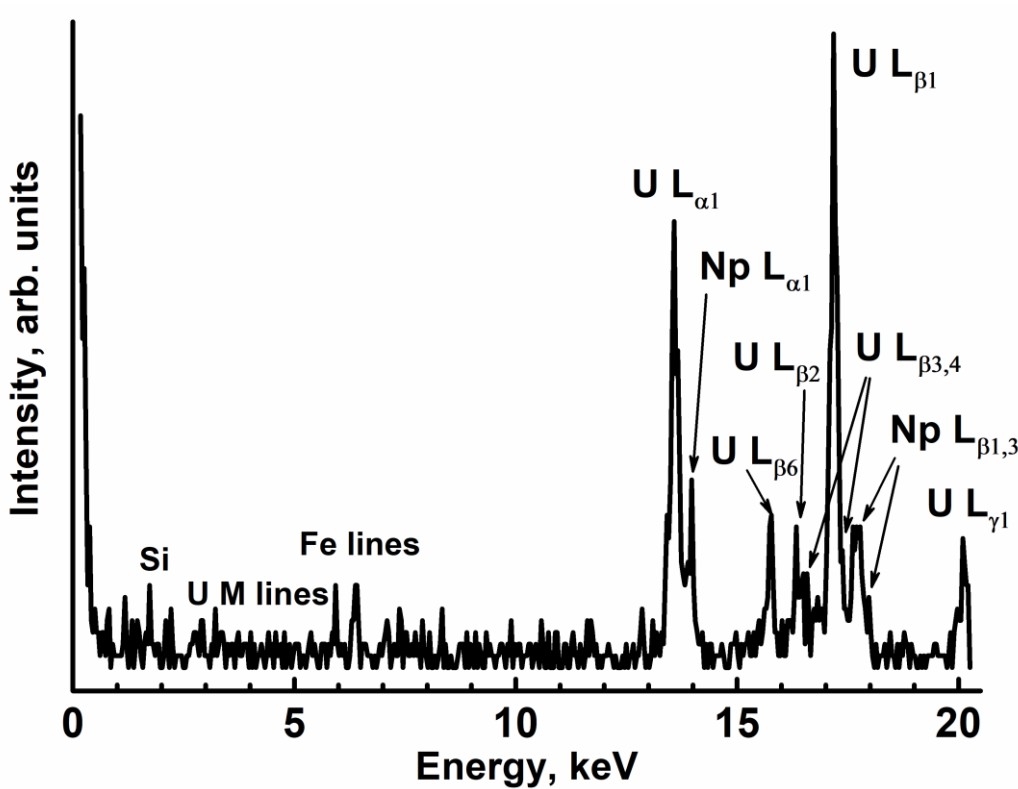

**Figure 3.** Autoemission spectrum of the sample R1 with tentative assignment of the peaks. The energy axis is binned down to 512 channels to improve statistics. Main peaks are assigned to most plausible transitions.

### 3.3. The Group I—Fully Vitrified Aerodynamic Fallout Particle (#R1)

Both halves of the glassy particle, R1a and R1b, demonstrate identical distribution of α- and β- radioactivity (Figure 4A–F). The highest density of α-tracks and the strongest signal on the imaging plate autoradiographs correspond to small glassy beads adhered to the particle surface. Several irregular, "smeared" domains in the particle bulk are also clearly visible on both types of autoradiographs. Comparison of the radiography with μ-XRF and SEM-EDX elemental mapping showed relative enrichment in Si and K and decrease of Ca, Al, Mg and Na in areas with high specific α-activity (Figure 4). At the same time, pure silica domains pronounced on Si maps and represented, most likely, by former quartz grains with diffuse boundaries, have the lowest content of radionuclides. Absence of measurable radioactivity in (former) quartz grains was also noted for products of underground GNOME explosion [38].

Analysis of composition and structure of the glass was carried out using SEM-EDX and Raman spectroscopy (Figure 4I). The sample R1 is almost entirely vitreous with several small relics of quartz grains and small amount of bubbles, mainly associated with the $SiO_2$ remnants. Weakly pronounced flow patterns, which might reflect spinning of the particle while in-flight, are observed. The majority of the cracks appear to involve inclusions of foreign phases and/or compositionally different domains, thus suggesting that differences in thermal expansion coefficient between the glass and former minerals were important for the crack initiation. Several compositionally different glass regions can be distinguished using the EDX data:

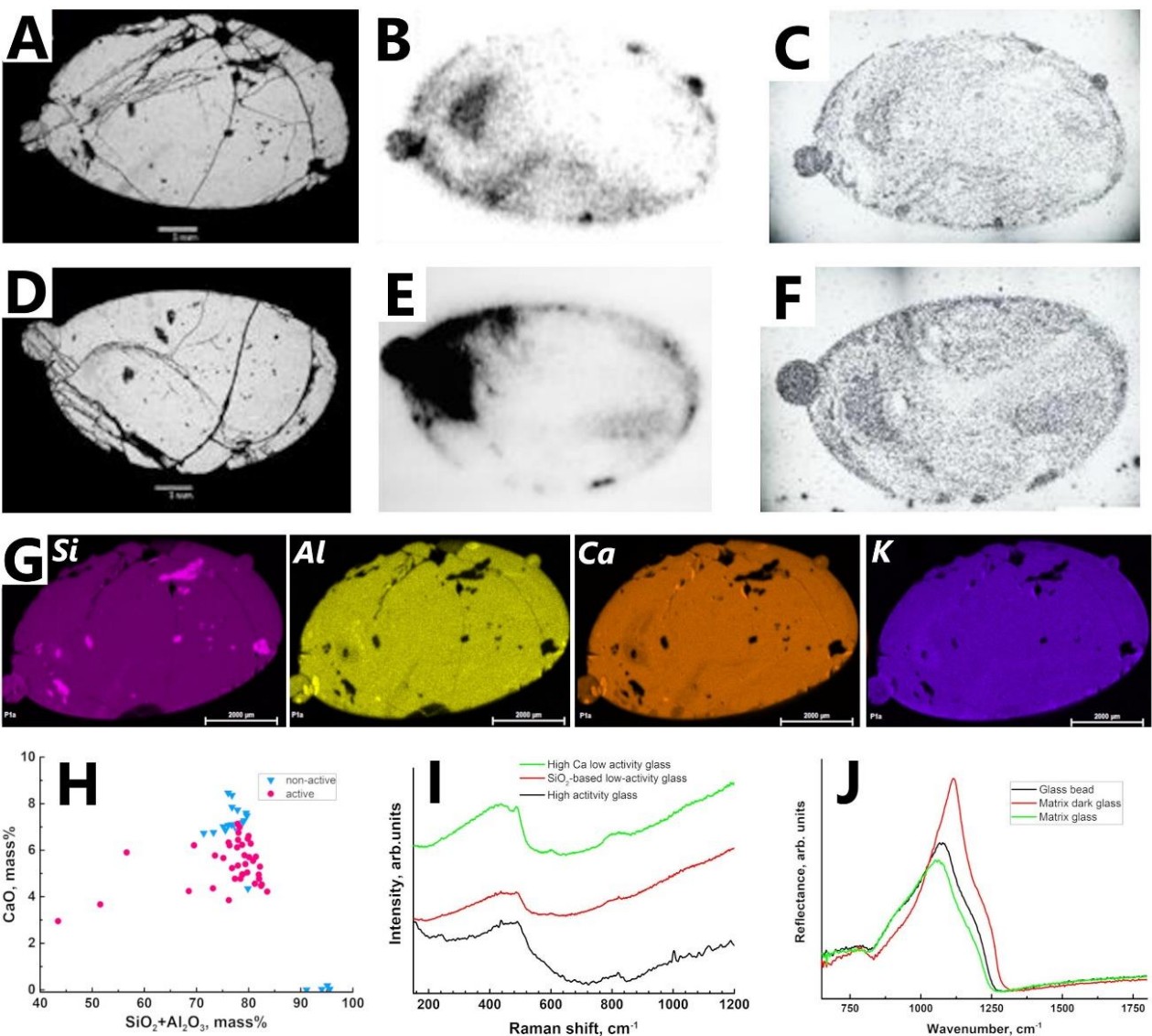

**Figure 4.** Structural and compositional peculiarities of both parts of the R1 aerodynamic fallout particle (R1a and R1b). (**A**,**D**) SEM-BSE images of R1a and R1b subsamples in epoxy; (**B**,**E**) α-track radiographs (19 h exposure); (**C**,**F**) imaging plate radiographs (2 h exposure); (**G**) micro-XRF maps of the R1a subsample: Si, Al, Ca, and K distribution; (**H**) compositional (CaO—($SiO_2$ + $Al_2O_3$)) plot for regions of the glass with contrasting radioactivity levels; (**I**) Raman spectra of low- (red and green curves) and high-activity (black) glasses of the R1b; (**J**) FTIR spectra of former quartz grains (red) and representative spectra of the glassy matrix.

(1)    The bulk of the particle is an aluminosilicate glass (60 mass% $SiO_2$; ~15–17% $Al_2O_3$; ~7% CaO; ~5% FeO; $Na_2O$~$K_2O$~MgO ~2%; $TiO_2$ < 1%);

(2)    Irregular patches of glass with wide variations in $Al_2O_3$ (8–17%), slightly poorer in MgO (~1.3–1.5 mass%), and CaO and slightly enriched in $Na_2O$ (~2.5%) relative to the bulk composition. The patches are present in the particle bulk and also may form embayments at the outer surface. In the latter case, the embayments might correspond to relics of dissolved surface beads described in point 5 below;

(3)    Remnants of quartz grains—spots of pure $SiO_2$ glass with diffuse boundaries;

(4)    Several $Al_2O_3$-rich spots with relatively high radioactivity;

(5)    Small glass beads attached to the main particle outer surface. The beads are slightly poorer in calcium (CaO 4.0—6.5 mass%), Na and K ($K_2O$~1.7%); totals of the analyses

are lower than those of the main particle matrix (~92–95%). These particles usually contain radionuclides.

The first two phases—the quartz relics and aluminosilicate glass—demonstrate similar Raman spectra (Figure 4I). The $Q^3$ structural units, typical for relatively polymerized (alumino)silicate glasses [39,40] dominate; strong luminescence contribution is observed. The reflectance FTIR spectra of the silica and of the aluminosilicate glasses in all other parts of the sample differ, reflecting compositional changes (Figure 4J). Note that in the FTIR measurements, the analyzed spot is much larger (50–100 μm) in comparison with the Raman one (few microns), thus providing spatially averaged information. The glass is anhydrous as indicated by absence of OH-related bands in FTIR spectra. The Raman spectra of the high radioactivity domain are rather weak but generally comparable with those described above. The peak at ~600 cm$^{-1}$ due to symmetric Si-O-Si bending vibrations in three-membered siloxane rings of "defectous" glass network [41] is less pronounced and is present as a shoulder. This behavior is unlikely to be induced by radiation damage from entrapped radionuclides due to their low absolute concentration. It is rather explained by deviations in chemical composition of the active domains from the bulk. Weak peaks of crystalline silicates (1000, 1112 cm$^{-1}$) are also present in most spectra.

Small beads on the surface of larger aerodynamic fallout particles are remarkable manifestations of turbulent airflows in the fireball, which led to collisions and merging of molten droplets, eventually, of diverse composition. Such merged objects are found not only in the fallout of near-ground tests, but also in underground confined explosions in salts, where they are present in bottom breccia [42], and in some rare varieties of tektites [43]. Interestingly, in the case of underground explosions, the smaller olivine beads may be completely engulfed by a carrier particle without obvious intermixing [42]. The interfaces between the host particle and small beads may preserve material directly exposed to the fireball environment from secondary alterations. The interface region may show a very complex structure, possibly reflecting kind of splashing during the merging. The interface thickness varies along the contact and shows some waviness in the thicker part. Our results generally confirm findings by Weisz and co-authors [8,23,24], revealing that the interface may be simultaneously enriched in Si and K and poor in Na, Ti, Mg, Al, and Ca. Decoupling of Na from K and coherent behavior of elements with markedly different volatilities remains unexplained and requires additional studies. Note, however, that not all beads show clear interface with the carrier particle. The latter beads might have merged with the carrier in early moments of the fireball development and had sufficient time for homogenization.

### 3.4. The Group II—"Khariton" Glass Particles

Particles of the second group abundant at Ground Zero possess a black surface color with a glassy luster. They are mostly elongated, irregularly shaped and reach 5–20 mm in the longest dimension (Figure 1G,H). According to X-Ray tomography data, these particles consist of a continuous outer glassy layer with rare inclusions of unmelted (presumably, relic quartz) grains, and a core composed of partly sintered but not molten sand with bubbles and cracks of various size (Figure 5A; Supplementary Video). The largest bubbles are concentrated at the interface with the glassy crust and were presumably trapped due to rapid quenching of the surface layer of the particles. SEM study of the polished cross-section confirms the tomography data: the interior of the particle is full of cracks and bubbles and has a loose structure while the outer shell is a dense structure and is polishable to a mirror finish (Figure 5B). Radioactivity distribution is highly heterogeneous; both α- and β-emitters are present exclusively in the outer glassy layer (Figure 5C,D).

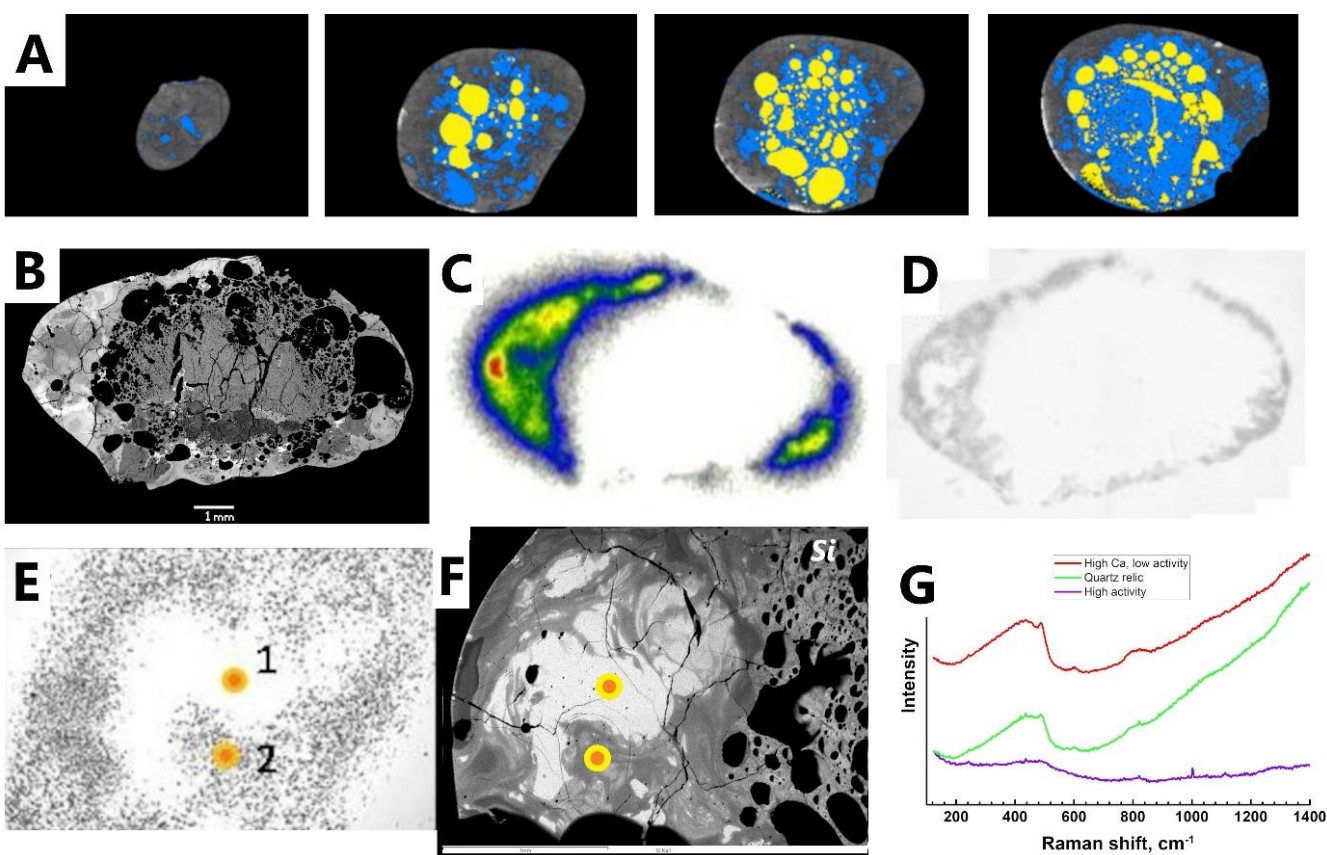

**Figure 5.** Structural features of the nuclear melt glassy fragments of Group II. (**A**) X-ray tomography slices of the particle Kh-2 with several phases of different density: glassy phase (grey), sintered sand (blue), and bubbles (yellow); (**B**) SEM-BSE image of the cross-section of the Kh-7b; (**C,D**) radiography of the cross-section of the Kh-7b: (**C**)—β-activity distribution on the imaging plate radiogram (6 days exposure); (**D**) α-track radiograph of the cross-section Kh-7b (6 days exposure); (**E,F**) a fragment of glassy shell of the particle Kh-7b: (**E**) α-track radiograph; (**F**)—Si map (SEM-EDX); (**G**) Raman spectra of points marked on (**E**) and (**F**).

### 3.4.1. Glassy Outer Shell

Microstructural and compositional peculiarities of the particle Kh-7 are presented in Figure 5. Highly active Al-Si glass dominates. Quartz-derived patches of pure silica glass, sometimes with traces of flow, do not contain α/β-emitting radionuclides (Figure 5E,F). The obvious reason for the sharp interface between non-active silica and the radioactive Al-Si glasses is the marked difference in melting temperatures (1670–1713 °C for quartz polymorphs and 850—1350 °C for aluminosilicate glass precursors such as feldspars) and rapid quenching, which prevented diffusion of radionuclides. A broad correlation between the alumina content and concentration of the α-emitters is observed. According to the Raman spectra, the $Q^3$ units dominate; the "defects"-related band at ~600 cm$^{-1}$ is present as a shoulder.

### 3.4.2. Iron-Rich Phases

Partial devitrification is observed in the internal part of the glassy shell adjacent to the particle core. Between the fragments of unmelted quartz grains, several generations of crystals with variable Fe-Al content derived from Fe-enriched aluminosilicate melt are present. The crystals of the first generation are enriched in Al and depleted in Fe, and a reverse compositional trend is observed in the second generation. The approximate ratio of Si:Al:Ca:Fe (at.%) in the residual glass is 14:9:5:7, in large crystals of the first generation: 16:13:6:2, and in small crystals of the second generation: 16:4:6:8. The Raman spectra

are dominated by magnetite peaks (Figure 6D). The precipitation of numerous poorly crystalline grains of sub-micron Fe oxides is plausible given high excess of Fe in the glass.

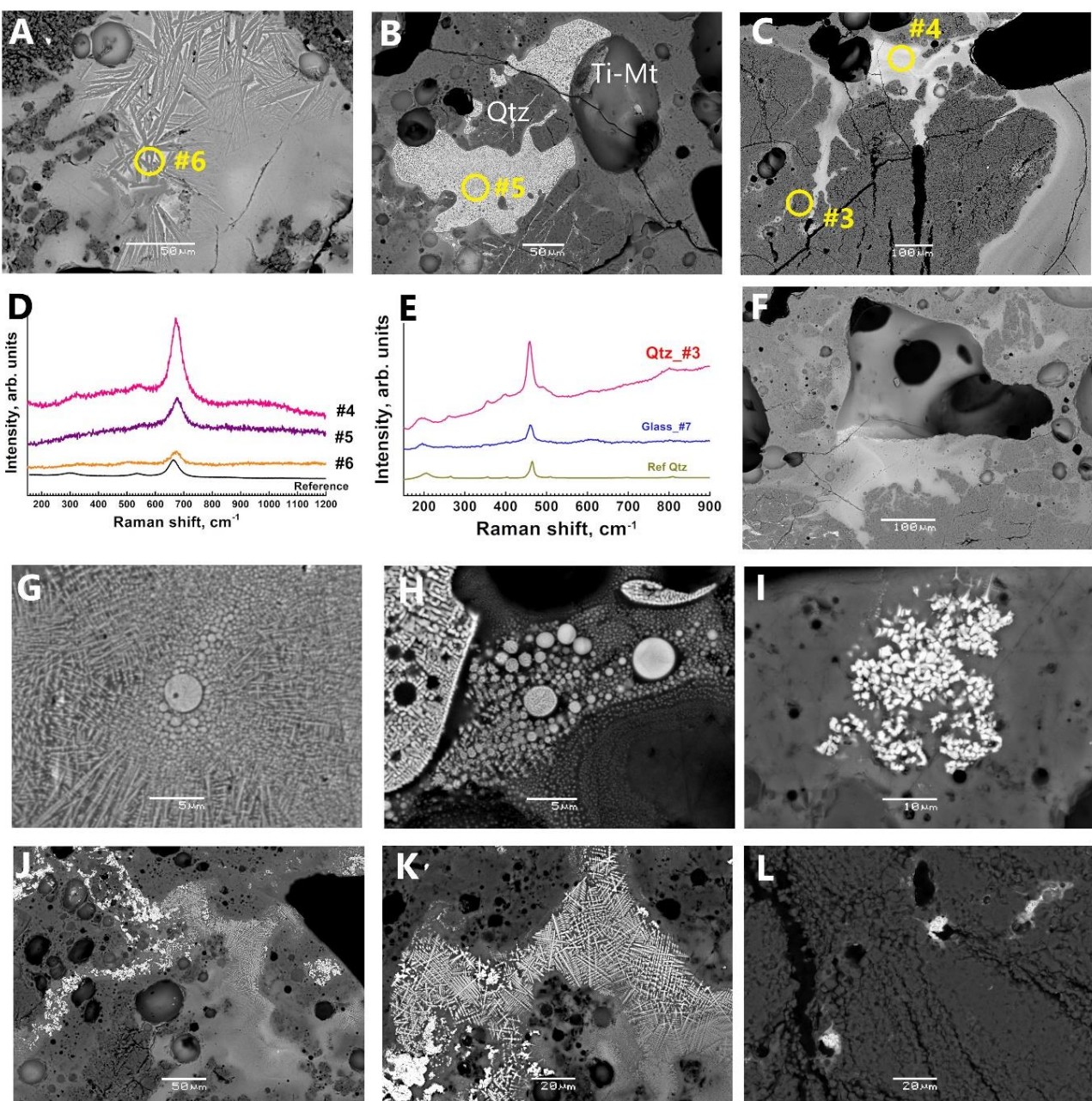

**Figure 6.** Structural peculiarities of the outer glassy shell of the Kh-7b sample. (**A–C,G–K**) SEM-BSE images of the Fe-enriched phases (white color) of different morphology; (**B,C**) crystal phases formed after incongruent melting of alkaline amphibole: Qtz—quartz, Mt—magnetite. Yellow circles show spots where Raman spectra (**D,E**) were recorded: #3—quartz grain (**C**); #4—Fe-enriched glass in a quartz crack (**C**); #5—Ti-magnetite (**B**); #6—magnetite in devitrified glass (**A**). (**D,E**) Raman scattering spectra of Fe-enriched phases (**D**), quartz grain and matrix of the aluminosilicate core (**E**). The references are from Rruff database [44]. (**F**) Interconnected porosity; (**G,H**) liquation texture; (**I**) pseudobrookite (Fe$_2$TiO$_5$) crystals in anorthite; (**J**) exsolution of Fe-oxides; (**K**) magnetite dendrites (zoomed region from (**J**)). (**L**) Sintered sand in the central part of the Kh-7 sample. White spots—former mineral inclusions.

A rather pronounced magnetite Raman peak was recorded in a Fe-rich crystallite, presumably formed from incongruently melted grain of an aluminosilicate mineral (Figure 6B,D).

Based on the presence of Zr and assuming loss of volatile Na and K, we believe that the precursor mineral was an alkali amphibole. Its melting produced a pure $SiO_2$ phase and Fe-enriched melt, which led to crystallization of magnetite, often with Ti admixture. Remarkably, in some spots, the magnetite aggregate follows the shape of a gas bubble (Figure 6B). Aggregates of Fe-rich crystals shown in Figure 6I are formed by pseudobrookite ($Fe_2TiO_5$) embedded into the anorthite matrix as shown by EBSD investigation. The presence of the pseudobrookite is an indication of high temperature–high $fO_2$ conditions. Liquation magnetite spheres with variable degree of internal structuring and sometimes surrounded by dendrites are observed in some regions (Figure 6G,H).

Interestingly, magnetite peaks are also observed in Raman spectrum of a glass caught in a crack of a large relic of quartz in the outer shell (Figure 6C,E). At the same time, individual crystalline phases are not apparent in the SEM-BSE images. This contradiction may be resolved in the assumption of high viscosity of the cooling Fe-rich melt, which precluded the formation of sufficiently large Fe-oxide crystals. High viscosity of the melt is apparent from examination of Figure 6C, where only limited penetration into a crack in a quartz grain have occurred. Not only individual bubbles, but also complex interconnected pores are encountered (Figure 6F).

The space between softened, but not yet molten quartz grains with diffuse boundaries, contains abundant bubbles and is filled with aluminosilicate glass with exsolved magnetite dendrites (Figure 6J,K). Presumably, the precursor minerals located between the quartz grains were characterized by variable (Si + Al + Ca)/Fe ratio. The average atomic ratio Fe/Ca of the phases are Fe oxide crystals—18/0.9, interstitial glass—6.5/1.5; liquation spheres—25/0.4. The mean composition of the Fe-enriched phase was recorded from a 1360 μm$^2$ area (at%): Si—13,5; Al—3; Na—1,5; Ca-0,9; Ti—1,5; Mn—0,4; Fe—18.

### 3.4.3. Internal Part: Glass and Sintered Sand

The core of the particle Kh-7b consists of a sintered sand, gradually evolving towards aluminosilicate glass with variable Fe, Ca, Na, and Mg content. Pure silica regions with sizes between $5 - n \times 100$ μm are scattered in the glass. Their Raman spectra are uniform throughout the core and are similar to the spectra of the silica glass in the outer layer. However, the position of the main quartz peak is downshifted from its reference position at 464 cm$^{-1}$ to 460 cm$^{-1}$ in the core and to 457–459 cm$^{-1}$ in quartz grains in the outer layer. Individual splashes of former barite ($T_{melting}$ = 1580 °C), monazite and zircon are scattered in the aluminosilicate matrix and in the glass (Figure 6L).

### 3.5. *Group III—Ground Glass*

Glass formed on the ground (Group III) comprises compositionally different layers, often preserving flow features. Quartz grains with variable degrees of amorphization and mechanical damage are the principal type of inclusions. Bubbles of markedly different sizes constitute up to half of the polished section. Distributions of α- and β-emitting radionuclides match each other and form irregular domains. The boundaries of radioactive zones on imaging plate radiographs are diffuse, since phases comprising β-emitters are macroscopic and are not infinitely thin. α-track radiographs are sharper and reveal not only distinct division of the sample into two layers with α-activities differing by at least two orders of magnitude, but also highlight details of the microscale distribution of α-emitting nuclides (Figure 7). In both zones, α-emitting nuclides are located in the glass matrix avoiding bubbles, quartz grains, and silica glass. The high-activity glass area in the sample HK-2A is generally enriched in Ca and Fe (Figure 7I). Raman scattering spectra of both types of glass correspond to moderately polymerized silica glass (Figure 7L). Elemental mapping of the heterogeneous area of the sample HK-1b proved that the high-activity glass contains more Si, K, and less Al, Ca, and Fe compared to neighboring areas of the low-activity glass.

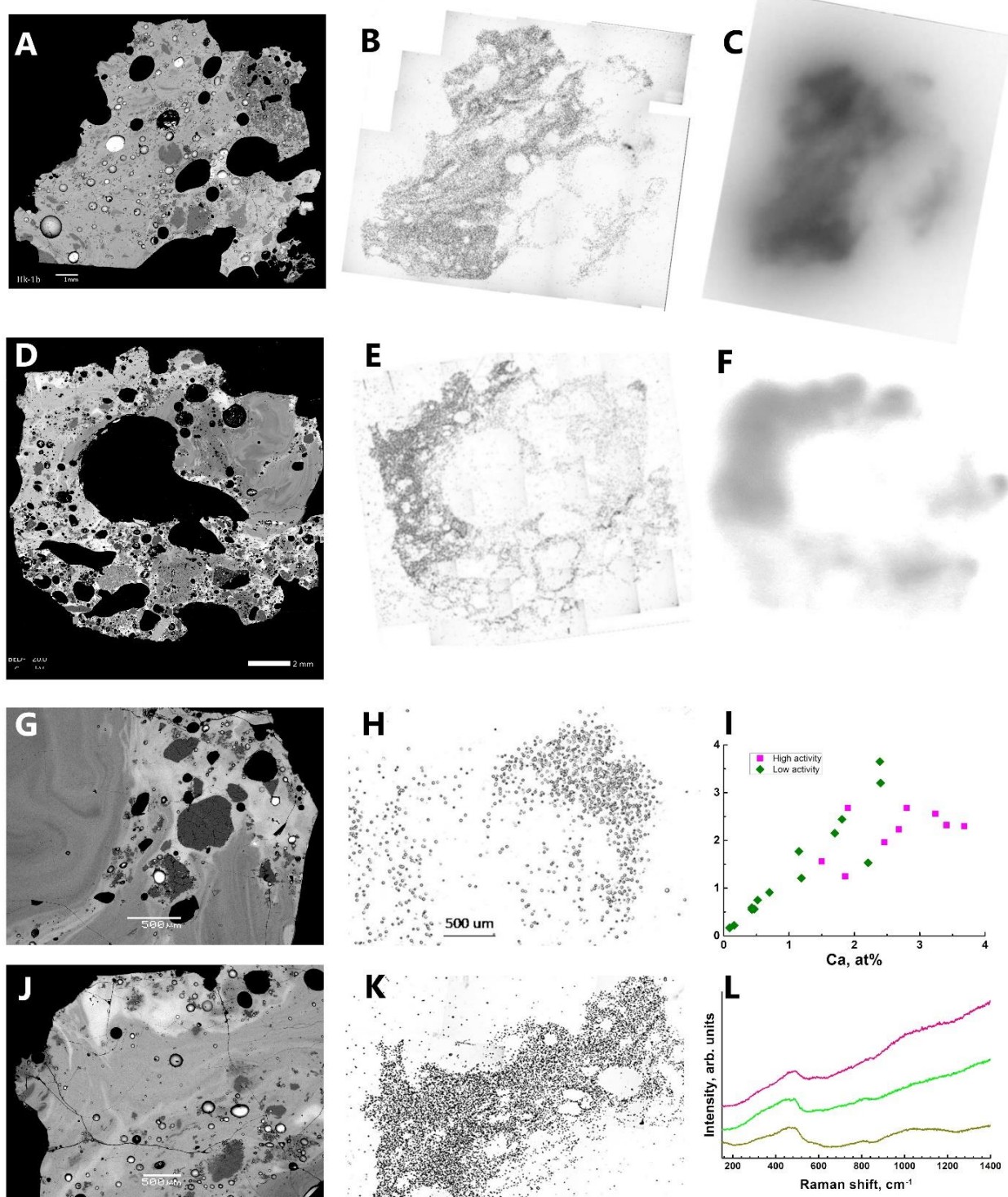

**Figure 7.** Ground glass particles HK-1b (**A**–**C**) and HK-2a (**D**–**L**); SEM-BSE (**A**,**D**,**G**,**J**), alpha track radiographs (exposure time: 6 days (**B**,**E**,**H**,**K**)), imaging plate (exposure time: 6 days (**C**,**F**)). SEM-BSE and alpha track images of the selected areas of HK-2a with low- and high-activity glasses (**G**,**H**,**J**,**K**) Raman scattering spectra (**L**) and Ca-Fe content in points with different activity (**I**).

Similar to some samples from Group II, processes of devitrification and mineral transformations reflecting melting and subsequent cooling with different rates are observed in the low-activity glass. The Raman spectra of the devitrified areas revealed the presence of magnetite (663, 560, 303 cm$^{-1}$) and anorthite (508, 974 cm$^{-1}$) (Figure 8A,B). According

to SEM-EDX and EBSD, the glass has decomposed to pseudobrookite (bright white in BSE mode), and anorthite crystals (approximate composition $CaAl_{2.7}Si_{4.1}O_{13.5}$). Recall that main peaks in Raman spectra of magnetite and pseudobrookite are similar.

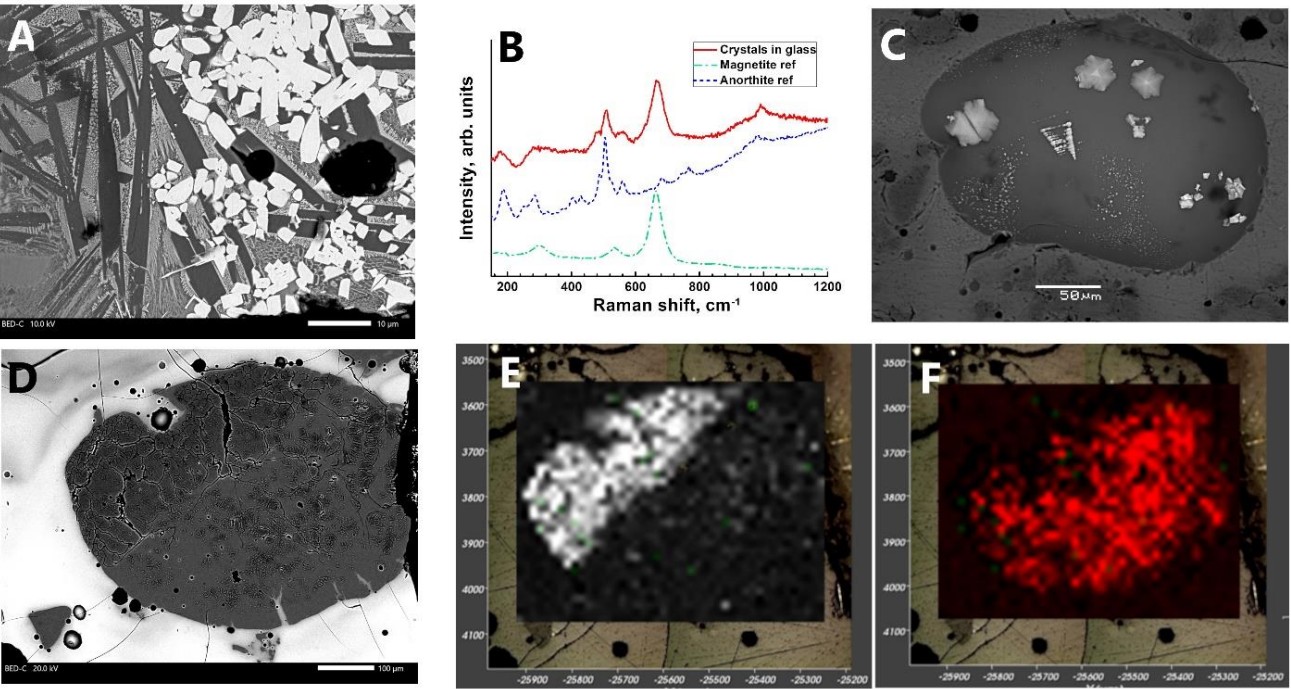

**Figure 8.** Peculiar mineral forms in the ground glass sample HK-2. (**A,B**) SEM image and Raman spectra of anorthite and Ti-magnetite; reference spectra are from [44]. (**C**) A bubble with Fe-oxide crystals decoration internal walls. (**D–F**) A quartz grain made of two $SiO_2$ polymorphs. (**D**) SEM-BSE image; (**E**) Raman map of cristobalite (mapped intensity in the 388–435 cm$^{-1}$ range); (**F**) Raman map of quartz (mapped range 442–476 cm$^{-1}$).

On walls of one of the bubbles exposed by cutting, crystals of a phase with obvious crystallographic faceting and dendrites are observed (Figure 8C). Exact identification is impossible due to the small size, but the most likely candidates are iron oxides. A possible explanation of their formation is based on localization on the bubble wall. Presumably, precursor Fe-oxide grains were touching the walls of the forming bubble and served as nucleation sites for FeO, which readily evaporates from aluminosilicate melt [45]. Such scenario implies temperatures above 1700–1800 °C, which is consistent with (partial) zircon decomposition, see part 3.5.2. below.

The presence of an unaltered chromite grain in the glass puts the upper limit on temperature experienced by the HK-1 sample to ~2200 °C.

### 3.5.1. SiO$_2$-Based Phases

The only crystalline phase in the high-activity glass is represented by non-radioactive $SiO_2$ grains with variable alteration degree. Most of these grains are covered by dense network of contraction cracks, but several unusual bi-phased grains are also observed, a representative one in shown in Figure 8B. SEM and Raman investigations show that it consists of two texturally distinct parts: approximately one third of the grain is heavily cracked, whereas the rest is largely intact (Figure 8D). According to Raman spectroscopy (Figure 8E), the cracked (upper-left on Figure 8D) part of the grain is α-cristobalite; the rest is quartz (Figure 8F). The presence of cristobalite indicates that the temperature locally exceeded 1470 °C, but pressure was less than 1 GPa. Small quartz crystals are scattered in both parts of the grain; judging from their shape and textural relationships it is likely that they represent shattered relic grains and not newly crystallized ones. Interestingly,

in samples produced during the "Trinity" nuclear test, the higher-temperature quartz polymorphs were not encountered. Unambiguous explanation of the crack pattern is barely possible, but it is clear that the grain has solidified while the surrounding material was still molten and able to penetrate into a crack visible in the lower part of the grain. The uneven cracking of the grain might be explained in assumption of relatively long persistence of a high temperature β-crystobalite polymorph in the cooling particle, until the matrix itself started to crack. When the matrix crack has reached the cristobalite grain, it served as a trigger for the high-to-low crystobalite transformation accompanied by volume effect.

In the inactive layer of the specimen HK-2A, a large quartz grain is enveloped by highly active glass (Figure 9A,B). A drop-shaped pore filled with silica and ore minerals: zircons, titanomagnetite, and iron oxides is found inside the quartz (Figure 9A,B). The presence of large amount of pores and marked enrichment of the grain rim and cracks directed towards surrounding glass implies evolution of a Fe-rich volatile component (Figure 9C). Exact identification of the precursor mineral is difficult, but iron-rich mica (e.g., annite) or feldspar are plausible. At the same time, the formation of nearly pure Ti-magnetite in place of former biotite was earlier observed in shocked rocks of Janisjarvi astrobleme experienced pressures exceeding 13–16 GPa [46]. In those rocks, isochemical transformation was excluded since non-shocked biotite from local shales does not contain appreciable Fe. It was suggested that interaction of the biotite with (volatile) decomposition products of adjacent minerals (e.g., quart and muscovite) could have markedly increased Ti and Fe content with a simultaneous drop in Si, Al, and Mg oxides in the final phase.

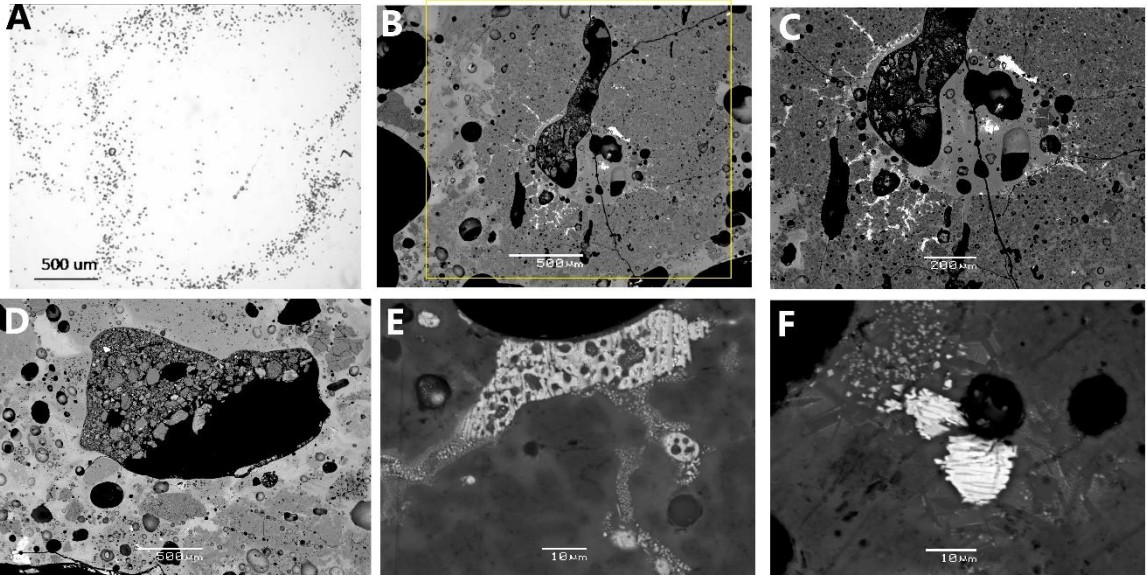

**Figure 9.** Pores and minerals in the ground glass sample HK-2a. (**A**) α-track image of a large quartz grain schematically marked by a yellow rectangle in SEM-BSE image (**B**). (**C**) Zoomed part of panel (**B**) showing abundant porosity of a metamorphosed mineral grain (see text for detail); white patches correspond to iron transport with expelled volatiles. (**D**) A pore filled with non-sintered material. (**E**) A destroyed ore mineral. (**F**) Ti-hematite intermixed with pseudobrookite; presumably, an ilmenite grain was a precursor.

A rounded mineral inclusion in the grain is represented by pseudobrookite intermixed with Ti-magnetite; presumably, this inclusion was an ilmenite. The presence of relatively loose mineral grains in the pores (e.g., Figure 9B–D) is a remarkable and unambiguous explanation why they are not fully molten and/or sintered is still absent. Presumably, high pressure of volatiles in the pore precluded the sintering. Such pores are encountered relatively often.

Chemical mapping (Figure 10) of two relic quartz grains revealed highly unusual distribution of alkalis. The alkalis (Na and K) form highly heterogeneous distribution patterns, consisting of tiny precipitates and a relatively wide diffuse band in the $SiO_2$ grain. At the same time, these elements are not correlated with aluminium, ruling out mechanical mixing of the former quartz with surrounding aluminosilicate melt. Another hypothesis consists of segregation of large low charge cations in a shock wave, an effect experimentally observed for Na/K in plagioclases [47]. Pronounced transport of alkalis is observed in plagioclase grains properly oriented relative to an intense shock wave. However, in our case, the K/Na-rich band follows the perimeter of the relic grain and lacks obvious direction. In addition, in plagioclases the segregation starts to be important at pressures above ~20 GPa, which is too high for the samples studied here. The most plausible scenario consists of the diffusion of alkalis into the quartz grain, possibly, assisted by high density of shock-induced defects. This model is indirectly supported by much more homogeneously and less abundant Na/K in a smaller grain in immediate vicinity. Presumably, the smaller grain contains smaller amount of defects, as also suggested by partial preservation of crystallographic faceting and relics of shattered quartz. A narrow, ~10 μm wide rim with low Na/K content in a larger grain is most likely an example of "uphill" diffusion of alkalis in silicate melt ([48] and references therein).

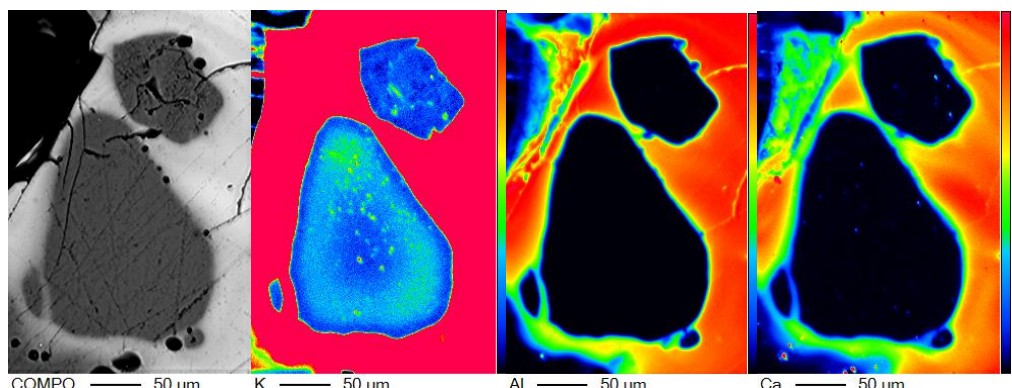

**Figure 10.** Chemical mapping of two adjacent relic quartz grains. Distribution of Na generally follows K and is decoupled from Al. See text for detail.

### 3.5.2. Zircon

Zircon grains show wide variations in degree of the decomposition (Figure 11). Many, but not all zircon crystals, show zirconia rims with vermicular structure (see also [49]). The temperature of thermal dissociation of $ZrSiO_4$ was assessed at $1673 \pm 10$ °C [50], although it strongly depends on grain size, defects, and annealing details. None of observed zircon grains was transformed to high pressure polymorph (reidite), implying a relatively low peak pressure. According to some studies, the morphology of replacing baddeleyite reflects the PT-conditions of the shock transformation of precursor zircon. Indeed, in a recent study of a fulgurite, a rather clear textural evolution of the baddeleyite phase with presumed peak temperature was noted [51]. However, it is clear, that not only temperature, but also composition of the surrounding medium will play an important role. For example, vermicular $ZrO_2$ rim was experimentally produced on a zircon crystal at just 1350 °C [52]. Influence of the composition and silica activity in surrounding melts of various viscosity is perfectly manifested in a markedly different extent of the zirconia rim development, ranging from the absence to a considerable degree on zircon grains separated by just 50 μm or even less (Figure 11A–C). The rim formation is markedly increased in the presence of a partly decomposed Ti-hematite grain; however, the composition of the surrounding glass appears to be rather constant within sensitivity of EDX method. Perfection of the zircon crystal lattice is another factor determining the decomposition onset and kinetics. As shown in Figure 11B,C, a relatively large zircon grain shows the preferential development of baddeleyite on a mechanically damaged surface and in some cracks. Since target soils

may contain zircon grains from different sources and ages, metamict or highly defectous crystals may coexist with (relatively) perfect ones. The decomposition rate of these grains to constituting oxides is expected to differ. Figure 11D,E show a fully decomposed zircon grain embedded into a relic quartz grain. This textural relationship between the host and inclusion markedly differs from most other zircon grains and might reflect different provenance, age, and lattice perfection.

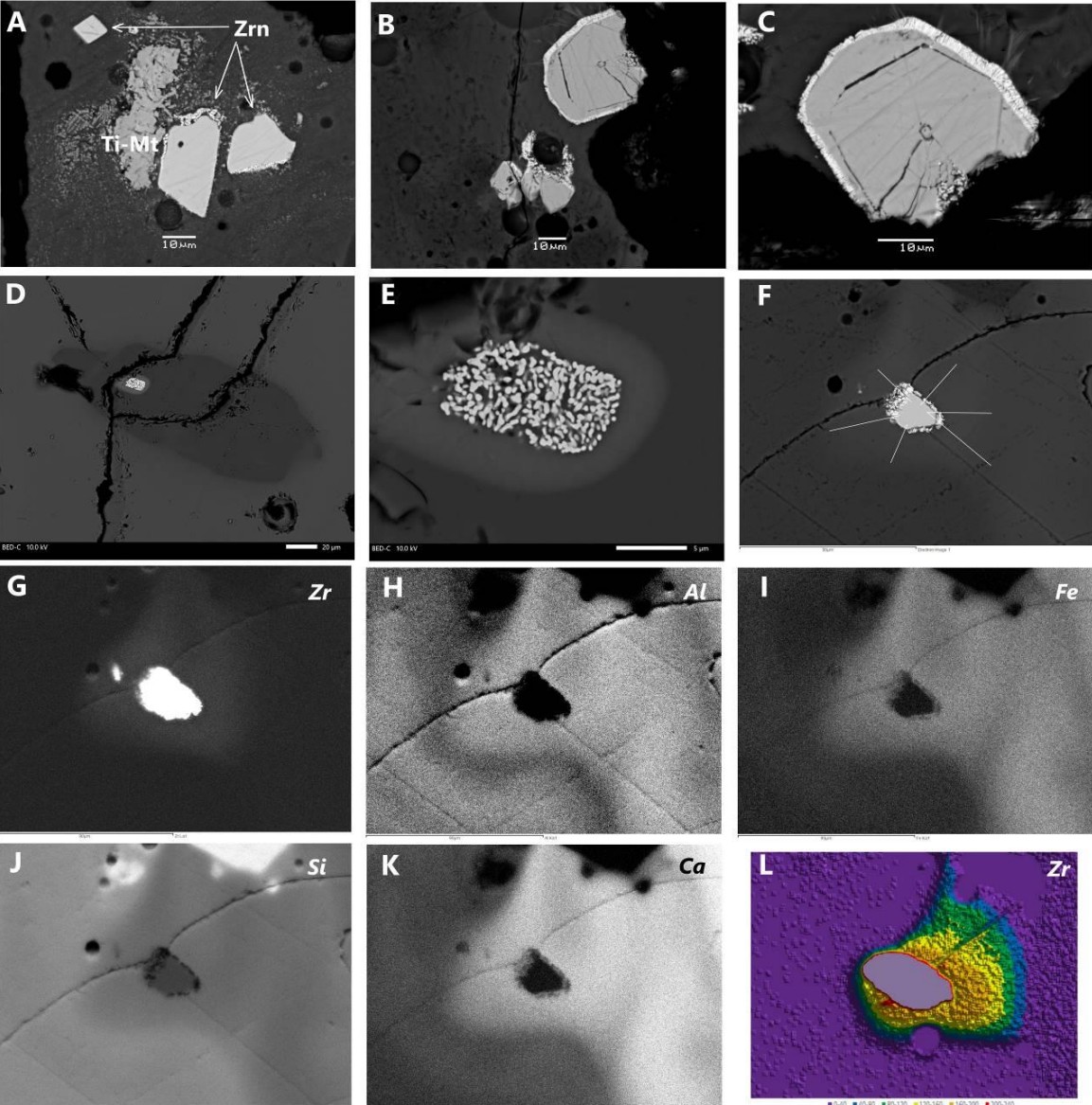

**Figure 11.** Zircon grains with different degree of decomposition into constituent oxides. (**A**,**B**) Adjacent grains ranging from perfectly facetted (upper left in (**A**)) to those with partial decomposition on the rim. (**C**) Zoomed grain from (**B**) showing decomposition of the cracked region (bottom right). (**D**,**E**) Fully decomposed grain residing inside a relic quartz grain (dark grey in (**D**)). (**F**–**L**) A partially decomposed grain. Direction of its motion in melt (see text for detail)—towards the bottom left corner of the image. Fine lines show directions of chemical profiles. (**G**–**K**) elemental maps ((**G**)—Zr, (**H**)—Al, (**I**)—Fe, (**J**)—Si, (**K**)—Ca). (**L**) 3D visualization of Zr distribution for the same grain after repolishing.

Distribution of zirconium and of several other elements in the glass matrix in regions adjacent to several partly transformed zircon grains was studied in detail. The grain imaged in Figure 11G–L shows variable degrees of decomposition on its facets; Zr distribution

in the surrounding glass is very uneven. This distribution might seem surprising, but examination of Zr profiles measured in several directions and chemical mapping suggest that the grain has started to decompose, but at a certain moment, it began to move in the surrounding melt, while the temperature remained sufficiently high for continuing decomposition. The reason for the displacement is unknown, but a plausible scenario implies either the falling of some object nearby or the onset of flow. These events are consistent with a model of ground glass formation from the precipitation of molten fallout. Interestingly, in all Zr profiles, a "notch" in the vicinity of the zircon face is present (see, e.g., the second experimental point in Figure 12D). The "notch" may correspond to the consumption of Zr by re-growth of zircon/badelleyite when the melt temperature dropped sufficiently. Note that in a "trinitite" specimen, a separation between $ZrO_2$ fibers and parent zircon was observed by TEM [49].

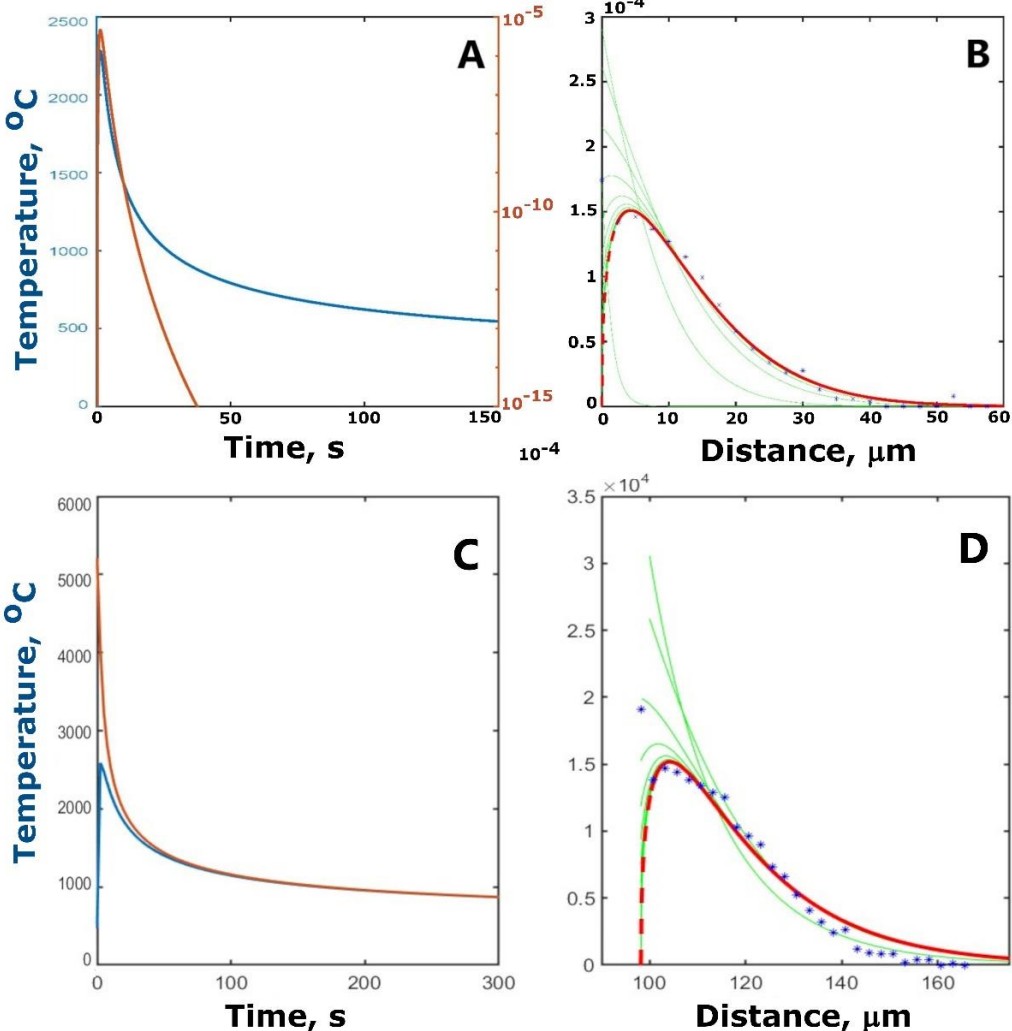

**Figure 12.** Time-temperature history of a zircon grain imaged in Figure 11F–L. (**A,B**) Calculation made for an averaged profile; (**C,D**) calculation for the "long" profile, see text for detail. Note markedly longer time and distance axis in (**C,D**) in comparison with (**A,B**). In (**A**) and (**C**): blue curves—best-fit temperature history for zircon dissolution; orange—corresponding variation of Zr diffusion coefficient. In (**B**) and (**D**) measured (stars) and simulated profiles of Zr concentration around the dissolving crystal (green thin lines). Quenched profile is shown by a red line.

In order to reconstruct time-temperature (t-T) evolution in the melt surrounding dissolving zircon crystal, we employed the model proposed in [53] for diffusive zircon growth in magmatic conditions. This model has been developed for a spherical grain and

neglects crystallographic faceting. However, in the studied samples, there is no obvious correlation between the faceting and the decomposition degree. Note also, that measured concentration profiles reflect projections of uneven volume distribution of elements on a plane exposed by the polishing. Nevertheless, we believe that comparison of results obtained for different profiles permits evaluation of the most plausible t-T path experienced by the specimen.

The transport of Zr in the silicate melt is governed by the diffusion equation that in a spherical case can be written as:

$$\frac{\partial C}{\partial t} = \frac{D(T)}{r^2} \frac{\partial}{\partial r}\left(r^2 \frac{\partial C}{\partial r}\right)$$
$$\ln(D) = -3.13 - 47000/T. \tag{1}$$

Here, $C$ is the concentration, measured in ppm, $r$ is the radius counted from the center of the zircon crystal, and $t$ is the time. Diffusion coefficient $D$ depends on the temperature that is assumed to be a function of time only, because the temperature equilibration timescale is several orders of magnitude shorter than the diffusive timescale. The Equation (1) is solved only inside the melt part of the cell $s < r < R$, where $s$ is the zircon surface and the outer radius $R$ is chosen to be much larger than the width of the dissolution zone to avoid the influence of far field boundary conditions.

Boundary conditions at the crystal-melt interface ($r = s$) assume local thermodynamic equilibrium and mass conservation of zirconium:

$$r = s: \; -D\frac{\partial C}{\partial r}\bigg|_{r=s} = J = V[C_m - C_z]; \; C_m = C_{sat}(T). \tag{2}$$

Here, $V$ is the growth/dissolution rate of zircon, $C_m$ is the concentration of Zr in the melt adjacent to the crystal, and $C_z$ = 490,000 ppm is Zr concentration in zircon. The saturation concentration $C_{sat}$ of Zr (Equation (3)) mainly depends on temperature and melt composition (M-factor, M = 1.3 for silicic melts [54]).

$$C_{sat} = 490,000 / \exp\left(\frac{10,108}{T} + 1.16(M - 1) - 1.48\right) \tag{3}$$

We specify the temperature variation with time using an analytical solution for slab cooling [55]:

$$T = \frac{(T_s - T_0)}{2}\left(\text{erf}\left(\frac{h - x}{\sqrt{kt}}\right) + \text{erf}\left(\frac{h + x}{\sqrt{kt}}\right)\right) \tag{4}$$

where $h$ is a slab thickness, $k$ is the thermal diffusivity coefficient, $x$ is the point where the temperature is evaluated, $T_s$ is the slab temperature, $T_0$ is the temperature for the enclosing media.

The temperature history of zircon grain shown in Figure 11F was calculated for several profiles. The solution is rather stable and Figure 12A,B shows results for the averaged profile. The longest profile is excluded from the averaging, since almost certainly it reflects two-stage dissolution (static and moving zircon, see above); its fit is shown in Figure 12C,D.

Following rapid increase in the temperature, the crystal starts to dissolve and zirconium diffuses out from the crystal-melt interface. As the temperature decreases, the equilibrium concentration drops down and the profile flattens. Because of the assumption of the local thermodynamic equilibrium, the model predicts rapid drop of the concentration at the crystal-melt interface. This drop is shown by a dashed line but does not match the observations due to failure of the employed model in highly non-equilibrium conditions. Due to low temperatures at the end of the process, the boundary condition on the crystal-melt interface does not influence the concentration profile at distances larger than 1 μm.

## 4. Discussion

Three groups of nuclear melt glass from the Semipalatinsk test site, different in morphology, origin, and radioactivity, have been studied using complementary analytical methods. From a radioecological point of view, the distribution and speciation of radionuclides is of the highest importance and is discussed below. Although the levels of radioactivity of the samples is well above detection limits of employed spectrometers and planar detectors, the absolute content of relevant chemical elements (Cs, Eu, Co, Am, Pu) are not measurable by EDX-spectroscopy. Consequently, their concentrations are governed not by equilibrium solubilities in a silicate melt, but rather by partition coefficient between the melt and vapor phase and speciation in dynamic conditions in the fireball.

Due to condensation inside the fireball, the aerodynamic fallout particles usually carry the highest amount of radionuclides [8,23]. This model explains the highest concentration of $\alpha$-emitting radionuclides, especially of $^{241}$Am, in samples from the Group I. At the same time, $^{60}$Co was not observed in these samples. The highest content of activation products $^{60}$Co and $^{152}$Eu is observed in ground glasses from the Group III. This is logically explained by intense neutron irradiation of soils and evaporated material of the tower. The "Khariton" glasses of the Group II specific activity of the activation and fission products decrease in line with $^{241}$Am.

The radioactivity distribution in all studied samples is highly heterogeneous. In the aerodynamic particles (Group I), the highest activity is associated with small, attached beads; in the Group II samples, the external crust is the most active. In the ground glass (the Group III) layers, the markedly different content of radionuclides are intermixed. Elemental composition of the most active glasses may differ both between the samples and within a single specimen. For example, in the crust of a single specimen HK-2 (the Group III) the domains with high Ca and Fe may contain markedly different amounts of $\alpha$-emitters (Figure 7). At the same time, in a similar specimen, HK-1b, the largest content of $\alpha$-emitters is observed in a Si/K-rich and Ca/Al/Fe-poor domain.

The chemical composition of the studied samples varies in a rather broad range (Figure 13). The largest scatter is observed for the ground glasses from the Group II, which likely reflects both involvement of soils from various locations and complex volatilization/condensation behavior in the fireball. In several studies on underground explosions [38,42], the highest concentration of radionuclides was observed in magnetite and other Fe-rich phases. In our samples, the correlation between the Fe-based minerals and radionuclides is absent. The difference between this work and [38,42] is explained by simple temperature-induced decomposition of precursor Fe-rich minerals in our case and gradual formation of the Fe-containing minerals during cooling of the explosive melt in underground tests. Note, however, that correlation between the Fe-rich phases and radioactivity may be present in some aerodynamic fallout particles, at least in the outermost layers [25].

Depending on its crystallographic perfection and thermochemical properties of the surrounding medium, a given mineral variety may show rather wide variations in temperature of decomposition. There is significant scatter in the decomposition degree of adjacent zircon grains; the markedly different extent of alkalis diffusion in closely situated quartz serves as a good examples of this behavior. Consequently, examination of a single phase may give misleading results. However, we were able to analyze several very different minerals, which allow the highest temperatures experienced by the ground glass to be constrained. Combined examination of zircon decomposition, destruction of barite, precipitation of Fe-oxides on pore walls (Figure 8C) and, finally, zoned crystobalite grains, imply that temperature of ground glass exceeded 1580–1670 °C. The upper boundary is set by perfectly preserved chromite grain, thus limiting the temperature to ~2200 °C. Duration of this heat pulse is difficult to estimate, although a completely independent estimate from the model describing zircon dissolution suggests that the flash was fairly short, possibly comparable with ~3 s duration in the Trinity test [2,3]. At the same time, the ground glass remained molten; possibly, the precipitation of still molten material from the fireball also

continued for appreciable periods. Our estimate on the duration of the molten state of the ground glass is based on modeling Zr diffusion profiles, which gives values up to ~40 s, with the exact value depending on the assumed melt solidification temperature. This relatively long period is indirectly supported by comparison of the modeled Zr profiles measured in different directions, which suggest that the examined zircon grain started to move 10–20 s after the onset of the decomposition (certainly, a very crude estimate). Retrospective analysis of Trinity test photographs suggested that 8–11 s after the detonation of the fireball separates from the ground and in-coming air masses quench the surface melt pool [56]. However, the fireball detachment does not necessarily lead to that rapid (not more than 2–3 s) melt solidification as assumed in [56], since the later depends on the melt thermal inertia and radiative properties and may vary.

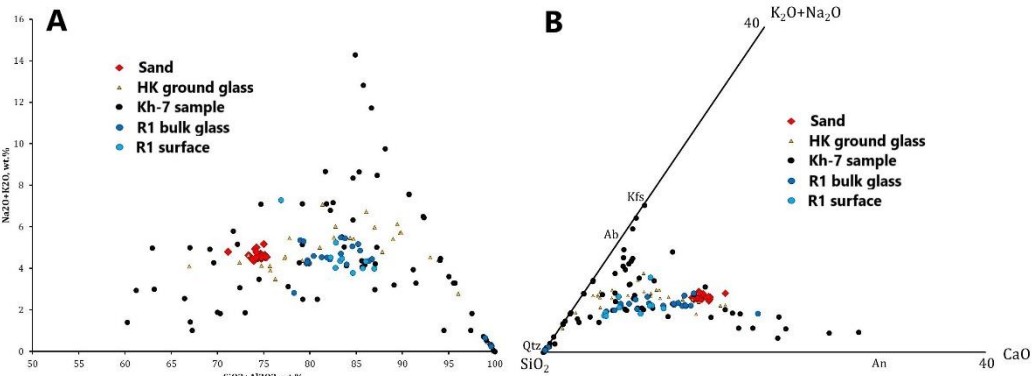

**Figure 13.** Chemical composition of samples from the studied groups in comparison with local sand. Representative points are shown. (**A**) part of a ternary diagram sum of alkalis ($Na_2O+K_2O$) – sum of $SiO_2$ and $Al_2O_3$. (**B**) $SiO_2$ corner of $SiO_2$-CaO-alkalis ternary.

The maximal pressure exerted by the blast on the soils appears to be rather limited. Whereas direct correlation of similarities between our study and biotite transformations in impacted rocks might suggest pressures in excess of 13–16 GPa, the presence of cristobalite implies much lower values, i.e., below 1 GPa. Absence of high pressure polymorphs also indicate moderate-to-low pressures. Similar conclusions were achieved for the Trinity test [4,49], although somewhat higher pressures are suggested. Subsequently, nuclear glasses differ considerably from impacted rocks, since they have experienced intense instant heating at a relatively low pressure. The cooling rate was likely much higher than in the impact craters due to the smaller volume of the melt pool.

Supposedly, fulgurites represent the closest natural analogue of nuclear melt glasses. However, widespread presence of highly reduced phases in fulgurites is a strong indicator of reducing conditions (e.g., [57]), presumably, assured by the reduction of very high electric currents. The nuclear melt glasses are, on average, more oxidized, although metal particles may be present in some varieties of ground glass [7] and aerodynamic fallout [14].

**Supplementary Materials:** The following supporting information can be downloaded at: https://www.mdpi.com/article/10.3390/en15239121/s1, Video S1: Sequence of slices obtained by reconstruction of X-ray tomographic projections of the particle Kh2 (Group II).

**Author Contributions:** Conceptualization, writing—I.E.V., V.O.Y., S.N.K., and A.A.S.; investigation—I.E.V., V.O.Y., A.A.A., R.A.S., T.R.P., I.M.N. and A.A.S.; software and modeling—O.E.M. and D.A.Z. All authors have read and agreed to the published version of the manuscript.

**Funding:** This research was funded by Russian Science Foundation, grant number 19-73-20051. SEM investigation and γ-spectrometry was supported by MSU development program. Spectroscopic measurements were performed using equipment of CKP FMI IPCE RAS. As part of X-ray tomography data image processing, the work was supported by the Ministry of Science and Higher Education within the State assignment FSRC «Crystallography and Photonics» RAS.

**Informed Consent Statement:** Not applicable.

**Data Availability Statement:** The data are available from the corresponding author on reasonable request.

**Acknowledgments:** We highly appreciate skillful sample preparation by Tolchinsky S.M. Some samples were provided by late Ogorodnikov B.I. We appreciate the useful comments of two anonymous reviewers.

**Conflicts of Interest:** The authors declare no conflict of interest.

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
