# Peer review of "Nuclear Melt Glass from Experimental Field, Semipalatinsk Test Site"

_energies, doi:10.3390/en15239121_

Round 1

Reviewer 1 Report

This is a very interesting manuscript with unique data. The manuscript is suitable for the journal and may be of interest to the reader. I recommend the article for publication.

Author Response

Dear Dr. Kullyakool

Please find attached a revised version of manuscript “Nuclear melt glass from Experimental Field, Semipalatinsk Test Site” by I.E. Vlasova and co-authors. We are grateful to reviewers for their attentive reading of our manuscript and useful comments. We have tried to address all of them. Suggested grammar corrections were implemented as well as number of other language amendments.

Replies to main comments from the Reviewer 1 are given below. Where applicable, additions in the revised manuscript are highlighted in yellow.

Comment How were particles found within background soils? Were they more radioactive than surrounding area (i.e., hot particles)? When were the particles collected?

Reply:

The particles were initially identified from their unusual appearance, markedly different from rocks common for the area: all the samples possess dark (black) glassy outer surface. All discussed samples are clearly considerably larger than local sand grains.

And yes, they can, to some extent, be referred as "hot" particles, since their radioactivity exceeds the radioactivity of the surrounding soils.

Comment The signification of the determined characterization could be more thoroughly discussed including the importance of particle size on transport and availability of the radionuclides present. A comparison to glass solubilities for the radionuclides detected would strengthen the discussion surrounding the sample spectrometry.

Reply:

The paper presents the results of a detailed study of individual particles. To make a sound discussion of the particle size effect on radionuclide transport, a large set of particles collected at well-defined different distances from the epicenter is necessary. Our sample set is far too small, moreover, there is a considerable uncertainty in assignment of a given specimen to particular test.

Regarding the solubility of radionuclides in the glass matrix: All detected radionuclides (cesium-137; europium-152; cobalt-60 and americium-241) have an extremely low concentration and are far from approaching the solubility capacity of glass. The following statement was added to Discussion section: Although the levels of radioactivity of the samples is well above detection limits of employed spectrometers and planar detectors, the absolute content of relevant chemical elements (Cs, Eu, Co, Am, Pu) are not measurable by EDX-spectroscopy. Consequently, their concentrations are governed not by equilibrium solubilities in a silicate melt, but rather by partition coefficient between the melt and vapour phase and speciation in dynamic conditions in the fireball.

Comment: …discussion into the leaching of the radionuclide from the glass matrix should be incorporated into the decay corrections to the explosion date.

Reply:

 Formed as a result of rapid cooling, the glassy particles are extremely resistant to both mechanical failure and dissolution, at least for several decades. No traces of leaching of radionuclides were registered. The following part of the manuscript is related to this issue:

The nuclear melt glasses are kinetically stable for several decades and effectively retain radionuclides: more than 30 years after the termination of the ground tests >90% of Pu and U in soils remain in acid-resistant residues [27-29]; the leach rates of actinides from glasses produced in underground tests at the STS are also low [30]. Since moratorium on ground and atmospheric tests effective since 1963, the major fraction of radioactivity in fallout plumes at the Experimental Field is concentrated in coarse soil fraction, largely in so-called «hot» particles [31].

Comment: the possibility of non-homogenous heating and pressure should be discussed. Namely, further discussion of the effect of the heterogenous melt on diffusion and crystallization is warranted.

Reply:

Of course, it would be interesting to discuss the issue of the melt heterogeneity on diffusion etc. This would require detailed chemical mapping of the whole particle with advanced modeling and is technically feasible, although very labor-consuming. However, it is not obvious that such modelling will give useful results, since we cannot really constrain peak temperatures and temperature evolution of different domains of the samples. A very interesting contribution by Weisz et al (ref. 24) shows results of relevant modelling and one can see huge variations in derived parameters depending on starting assumptions.

Comment: To facilitate such discussions within the current breakdown of the results section, a combined results/discussion should be considered. This would allow the authors to frame the significance of the particle characterization results while avoiding redundances in the discussion section. A conclusions section that provides further perspective on the overall significance of the present study to the overall community beyond detonations would strengthen the article.

Reply:

Indeed, such changes could be implemented, although their impact is uncertain. The principal problem is a very limited time given for the revision.

We hope that the revised version will be suitable for publication.

On behalf of all coauthors,

A.A. Shiryaev

Reviewer 2 Report

Article provides a detailed characterization of 3 types of particles from the Semipalatinsk test site. Particles have three different origins/profiles from within the blast(s): spherical aerodynamic glassy fallout particles, elongated/irregular shaped aerodynamic particles, and nuclear melt ground glass. Each particle type is well characterized and provides important insight into the material produced during a nuclear blast/incident. The discussion and modeling of zircon grain decomposition provides important analysis of blast temperature influence on particle production. Overall, the article is novel but would benefit from additional detail to better fit the scope of the journal. With these editorial and organization changes, the article is fit for acceptance for publication.  

The introduction provides an appropriate amount of background information. Characterization methods are well stated but more detail on the identification and collection of the radioactive materials would broaden the audience of the article, more directly relating the article to meltdown and other high temperature nuclear incidents beyond detonations. For instance: How were particles found within background soils? Were they more radioactive than surrounding area (i.e., hot particles)? When were the particles collected? The Results section provides an appropriate level of detail on the characterization results. The signification of the determined characterization could be more thoroughly discussed including the importance of particle size on transport and availability of the radionuclides present. A comparison to glass solubilities for the radionuclides detected would strengthen the discussion surrounding the sample spectrometry. Also, a discussion into the leaching of the radionuclide from the glass matrix should be incorporated into the decay corrections to the explosion date. The significance of particles still being intact and the possible change in crystallization after years of environmental exposure (if applicable – see comments on methods) could strengthen the discussion. Finally, the possibility of non-homogenous heating and pressure should be discussed. Namely, further discussion of the effect of the heterogenous melt on diffusion and crystallization is warranted.

To facilitate such discussions within the current breakdown of the results section, a combined results/discussion should be considered. This would allow the authors to frame the significance of the particle characterization results while avoiding redundances in the discussion section. A conclusions section that provides further perspective on the overall significance of the present study to the overall community beyond detonations would strengthen the article.

Specific Comments

L18 – provide[s]

L25 to L26 – sentence is misleading as radionuclides are found in the glassy phase (s). Consider revision

L41 - nuclear weapon[s]

L44 – comprise [of]

L51 to L55 – run-on sentence

L68 – remove “or”

L133 – [was] used

L193 – [The] low value

L239 – Provide reference to section where element concentration are discussed

Figure 3 – Ensure font ULB2 and ULB6 peaks is different from the rest.

Figure 4 – consistency with legend outline  

L316 – remove eventually (redundant)

L383 – follows [the] shape

L433 – provide more detail on meaning of “uneven” in this context

L490 – consider rewording “uneasy to explain”

L504 – remove possibility (redundant)

L570 to L572 – revise sentence, improper use of semicolon

L573 – Figure (not figures)

L619 – define s

Figure 12 – axis should have consistent fonts

Author Response

(The authors gave the same response as above.)
